# Protocol for a randomized clinical trial comparing the efficacy of Structured Diet (SD) and Regular Therapy (RT) for adolescents with malnutrition having Autism Spectrum Disorder (ASD)

Ruksana Akter[1], Nadia Afrin Urme[2], K. M. Amran Hossain[3], Tofajjal Hossain[3], Shameem Ahammad[4], Mst. Hosneara Yeasmin[5], Md. Zahid Hossain[3], Rashida Parvin[1], Md. Sohrab Hossain[2], Md. Ashrafuzzaman Zahid [1]*

1 Department of Nutrition & Food Technology, Jashore University of Science & Technology, Jashore, Bangladesh, 2 Department of Physiotherapy, Bangladesh Health Professions Institute, Savar, Bangladesh, 3 Department of Physiotherapy & Rehabilitation, Jashore University of Science & Technology, Jashore, Bangladesh, 4 Department of Occupational Therapy, Mymensingh College of Physiotherapy and Health Sciences, Mymensingh, Bangladesh, 5 Department of Speech and Language Therapy, Centre for the Rehabilitation of the Paralysed (CRP), Savar, Bangladesh

* ashraf@just.edu.bd

**Data Availability Statement:** No datasets were generated or analysed during the current study. All

# Abstract

## Background

Autism spectrum disorders (ASD) have a lifelong impact on behavior, communication, cognitive function, education, physical functioning, and personal, or social life. Separate studies suggest, Therapeutic and dietary interventions are effective to some extent in managing these issues. No study integrated the nutrition and therapeutic approaches and examined the outcome on disease severity, overall health, and behavioral status in ASD. The proposed study is designed to evaluate the combined effect of regular therapy (RT) and structured diet (SD) compared to the usual diet (UD) for Adolescents with ASD.

## Methods

The proposed study will be a randomized clinical trial (RCT) with the assessor, therapist, and participants blinded to group allocation. Seventy ASD children with malnutrition will be enrolled in two different facilities of the Centre for the Rehabilitation of the Paralysed (CRP) between January 2023 and June 2023. Participants will be enrolled through a hospital-based randomization process from a population-based screening dataset, and with a concealed group allocation to either RT+ SD or RT+ UD group with a 1:1 ratio. The outcome measures are the Childhood Autism Rating Scale as per DSM-5 to determine the severity of ASD, Mid-upper arm circumference (MUAC), and BMI for nutritional status, and Gilliam Autism Rating Scale (GARS-2) to assess the behavioral status. Post-test will be performed after 12 weeks of intervention, and Follow-up will be taken after 6 months of post-test.

relevant data from this study will be made available upon study completion.

**Funding:** Partial funding of the trial was funded by the Bela Health and Education Foundation (BHEF), Grant number BHEF/2022/9/G2. The fund was specified for the identification and intervention provision for the Adolescents with Autism Spectrum Disorder in Bangladesh. Publication or any other costs are self-funded by the authors.

**Competing interests:** The authors have declared that no competing interests exist.

## Perspectives

The result of the study will contribute to the provision of a comprehensive approach to malnourished Adolescents with ASD, and manage the issues related to the severity of ASD, stereotypical behavior, and anticipated health hazards.

**Clinical trial identifier:** CTRI/2022/11/047653.

## Introduction

Autism is one of the leading neuro-developmental disorders in the global context that interferes with social communication and basic conduct [1]. Although it can be diagnosed at any age, the manifestations of ASD are often delivered throughout the first two years of life [1, 2]. Autism Spectrum Disorder (ASD) affects people of all ethnic, racial, and socioeconomic backgrounds [1–3]. Children with ASD experience a variety of social problems, and repetitive or restricting behavior that is commonly observed during interaction, language, and communication [4, 5]. In Bangladesh, the prevalence of ASD has been steadily growing [6]. However, there is no epidemiological evidence of the overall number of autism cases in Bangladesh. Data from the Bangabandhu Sheikh Mujib Medical University's Center for Neurodevelopment and Autism in Children suggest that 7.5 per 10,000 adolescents [7]. The study suggests, a larger proportion of children with ASD in urban areas seek treatment from a tertiary facility named the Institute of Pediatric Neuro Disorders and Autism (IPNA). National-level research on 7200 people conducted in Bangladesh in 2013 discovered a prevalence of ASD of 1.5 in 1000 children [8]. The rate was higher in Dhaka City (30 out of 10,000 children), compared to rural areas (7 out of 10,000 children) [7, 8].

Children's growth, development, and nutritional condition are valuable indicators for determining children's health [9]. Children with ASD may experience issues related to inadequate nutrition, such as limited eating behavior, appetite suppression, or metabolic issues leading to inadequate growth [10]. A study in Bangladesh found children with ASD had significantly lower height and weight, wider triceps and supra-iliac skin fold, and were underweight in comparison to non-ASD children [11]. The reason might be, they are often neglected at the dietary level due to the lack of awareness and non-adherence to evidence-based interventions. There is no structured dietary protocol for children and adolescents with ASD in Bangladesh. Global researchers from different countries found a positive effect of structured dietary protocols for children with ASD [12]. Improper diet can lead to gastrointestinal manifestations including inflammatory bowel disease, food sensitivity and allergies, infections, and illnesses associated with biological and viral infections [13].

There have been several systematic reviews and meta-analyses [14–16] conducted on the nutrition of ASD children. There are different opinions on dietary interventions for ASD in Adolescents [12, 17, 18]. It is assumed that gluten-free and casein-free diets can minimize the symptoms of ASD, but there are controversial theories about the effectiveness of the gluten-free and casein-free (GFCF) diet [17]. According to certain studies, the GFCF diet can assist with symptoms such as stereotyped attitude, cognitive and social function, attention, interaction, and mood [18]. Many explanations have been proposed to explain how the GFCF diet improves the features of autism. Of all of these, the opioid excess hypothesis is the most widely accepted. Gluten and casein both affect the central nervous system via the leaky gut, but opioids escape via inflamed and thinning stomach in children with ASD [12].

Therapeutic interventions are multidisciplinary approaches [19] that, composed of balance [20], gait training [21], Therapeutic horse riding [22], motor training [23], strength training [24], aerobic exercise [25], and executive functional tasks [26]. It is hypothesized that structured dietary intervention adjunct with regular treatment approaches may have a greater benefit in children with ASD but no such research in Bangladesh has been held yet. It is important to determine the combined outcome of a structured diet (SD) and regular therapy (RT) for improving nutritional status and behavioral indices in the child and adolescents with malnutrition having Autism in Bangladesh.

The study objectives are to (1) determine the socio-demographics and anthropometric measures related to malnutrition for ASD in Bangladesh, (2) to elicit the within-group difference between structured diet (SD) with regular therapy (RT) or usual diet (UD) with regular therapy (RT) therapy on nutritional status and behavioral indices in post-test (3months) and follow-up (6months) compared to baseline, and (3) to find out the effectiveness among groups (SD with RT versus UD with RT) and observations (pretest to post-treatment, pretest to follow up, and post-test to follow up) comparison in the improvement of nutritional status and behavioral indices.

## Method

### Study design

The study will be a randomized clinical trial (RCT) with the assessor, therapist, and participants blinded to group allocation. ASD adolescents having malnutrition will be enrolled in two different facilities of the Centre for the Rehabilitation of the Paralysed (CRP) from January 2023 to June 2023. Participants will be enrolled through a screening process through a population-based cross-sectional household survey. Population-to-sample frame determination will be proceeded by stratified random sampling, sample enrollment in the trial will be performed by a hospital-based random sampling process and group allocation will be performed using concealed allocation. Both groups will have a similar number of participants (1:1).

To ensure the quality of the interventional study, we will follow Standard Protocol Items: Interventional Trials 2013 (SPIRIT) criteria for this experiment (Fig 1).

### Sample size

Sample size calculation has been performed through the software ClinCalc [S1 File] estimating the key outcome as the score of the Gilliam Autism Rating Scale (GARS 2) second edition [23, 30]. Sample size has been calculated [27] as the anticipated minimal clinically important differences (MCID) of GARS-2 [23] were estimated as 4.7±1.7 (0–10 Hiva scale converted from 0–126 GARS 2 score) with a baseline of 25% minimal clinical improvement, enrolment ratio 1:1, 80% power, and with the alpha value 0.05, the total sample stands as of 66. For safety we will enroll 70 ASD adolescents with a number of 35 participants in each group.

### Recruitment and randomization

The researcher will perform a sample frame to pool the samples for the interventional design. The sample frame will be taken from the dataset of the list of maltreated ASD Adolescents of Bangladesh screened by an early population-based household survey. From the sample frame, Participants living in Dhaka will be contacted one by one and will be invited to take part in this trial, thus hospital based-randomization will be ensured. After the pooled sample estimation, concealed allocation to groups will be performed using the "rand" function in Microsoft Excel 2010. Both groups will be treated in two different centers of an organization to ensure an

| Time point | Enrolment | Allocation | Post-allocation | | |
|---|---|---|---|---|---|
| | -$T_1$ | $T_0$ | $T_1$ | $T_2$ | $T_3$ |
| **Enrolment** | | | | | |
| Eligibility screen | X | | | | |
| Informed consent | | X | | | |
| Demographic assessment | | | X | | |
| Group allocation | | X | | | |
| **Intervention** | | | | | |
| PT | | | | X | |
| OT | | | | X | |
| SLT | | | | X | |
| Dietary Advice | | | | X | |
| **Assessment** | | | | | |
| AM | | | ◆————————————◆ | | |
| GARS-2 | | | ◆————————————◆ | | |

*PT=Physiotherapy; OT= Occupational therapy; SLT= Speech and language therapy; AM= Anthropometric Measurement; GARS-2= Gilliam Autism Rating Scale; $T_0$=Group allocation; $T_1$=Baseline before the intervention; $T_2$=Measurement taken in 3-months after $T_1$; $T_3$= Measurement taken after 6-months of $T_2$.*

**Fig 1. Standard protocol items: Interventional trials 2013 (SPIRIT).**

equal level of care is being provided and prevention of cross-contamination of data. In the group allocation process, block randomization will be employed to ensure the balanced recruitment for two groups.

## Study procedure

After the trial registration is obtained, researchers will start the phone call procedure to estimate the willing participants. As the primary screening of cases of "ASD with malnutrition" performed in the population-based survey, the parents will be directly contacted for participating in the trial. From the computer-based random serial, the first 70 ASD will be pooled for enrollment in the study. On the first day, an assessor blinded to group allocation will further screen and take a pre-test assessment. Then the participant will be treated in a multi-disciplinary approach with the consultation of a nutritionist, physiotherapist (PT), Occupational therapist (OT), and speech and language therapist (SLT). Both structured dietary treatment and regular therapy will be provided as three 45 minutes sessions per week for a total of 12 weeks. After 12 weeks, the post-test data will be obtained by the assessor. There will be a follow-up evaluation after 6 months of post-test. There will be a separate assessor for each of the treatment set-ups. The Consolidated Standards of Reporting Trials (CONSORT) manual will be followed when conducting the study. Fig 2 provides a summary of the study's methodology.

## Study setting

Both of the groups will receive treatment from two different centers of the Centre for the Rehabilitation of the Paralysed (CRP), the largest rehabilitation facility in Bangladesh. Specification

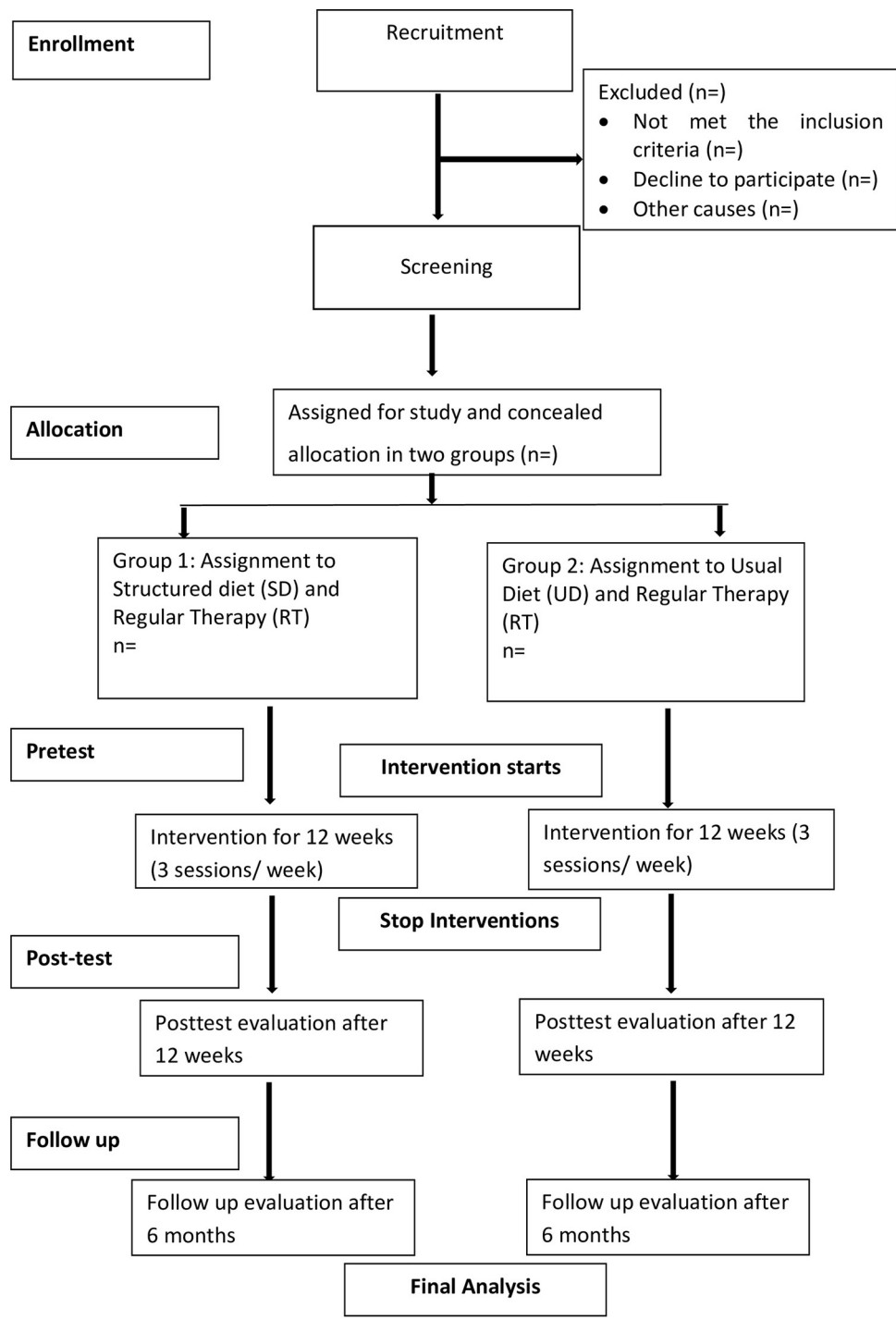

**Fig 2. CONSORT flow chart of the proposed trial.**

of centers for therapy group allocation will not disclose, to ensure the physiotherapists and the patients are blinded to group allocation. Two study centers of an organization (one for control and one for experimental) were selected because of the availability of specialist multi-disciplinary team of physiotherapist, occupational therapist, speech and language therapist, child

psychologist, educational psychologist, nutritionist and special educators were available alto-gether in that specialized organization.

## Eligibility criteria

Participants will be recruited if (1) aged between 13 to 18 years, (2) diagnosed as ASD children based on DSM-V criteria [28], (3) malnutrition according to Body Mass Index (BMI) [29], (4) not taking any diet advice or supplement, and (5) willing to participate. On the other hand, participants will be excluded if- (1) respondents provided consent but dropped out within the first week of enrollment, (2) allergy to any gluten-free, casein-free (GFCF), Soya free diet, (3) severe behavioral problems according to Gilliam Autism Rating Scale (GARS-2) [30].

## Outcome measurements

**Explanatory variables.** *Socio-demographic information*. Socio-demographic information will be recorded as participants' personal, demographics, and other disease-related informa-tion by a structured questionnaire prepared by the researcher.

*Diagnosis of ASD*. Screening of Autism spectrum disorder (ASD) children will be based on the planned diagnostic criteria of the DSM-5 [28] committee. ICD-10 and DSM-IV-TR criteria for ASD were significantly revised in DSM-5 [31]. In the DSM-5, four previous types of Autism in DSM-IV (Autistic disorder, Asperger syndrome, childhood disintegrative disorder and pervasive developmental disorder- not otherwise specified (PDD-NOS) was merged in an umbrella term Autism spectrum disorder (ASD). Also, the first diagnostic category of ASD indicates deficits in social communication and interaction. For an ASD diagnosis, all of the fol-lowing symptoms must be present (1) impairments in socio-emotional engagement with another person, (2) problems in nonverbal behaviors during communication, (3) problems in making or continuing communication with others. The presence of two or more of the follow-ing symptoms associated with limited interest and repetitive behaviors is required for the sec-ond domain as (1) persistent or repetitive speech, movement, or holding an object, (2) adherence to routines, rituals of verbal or nonverbal behavior patterns, or excessive resistance to change, (3) abnormal fixated interests or intensity of focus, (4) more or less reactivity to sen-sory input, or (5) unusual interest in sensory aspects of the environment. According to the third category, symptoms must appear in infancy but may not completely develop until socie-tal demands exceed capacity. The symptoms must interfere with daily functioning. Also, the diagnosing psychologist offers severity ratings as social communication, limited interests, and repetitive patterns of conduct, ranging from 1 to 3. The highest severity grade of 3, denotes that the domain is seriously impairing users [32]. This tool shows approximately 98.5% sensi-tivity and 92% of specificity to diagnose ASD [33].

**Primary outcome variables.** *Gilliam Autism Rating Scale (GARS-2)*. The second edition of the Gilliam Autism Rating Scale (GARS-2) will use to assess the behavioral status and overall status of ASD Adolescents. It is a behavioral screening tool for individuals with ASD between 3–22 years of age, based on DSM-IV (Diagnostic and Statistical manual of mental disorder) criteria for ASD. This scale comprises 42 items ranging from 0–126 scores, divided into 3 sub-scales (stereotyped behavior, communication, and social behavior) [30].

*Anthropometric measurements (AM)*. Anthropometric measurement (AM) will use to mea-sure the nutritional status of the child. Because it is non-invasive and less expensive than other methods, anthropometry, and circumference of the arm, abdomen, and thigh (MUAC) have long been considered nutritional status determiners [34]. BMI [29] of the child and adolescent will be measured as another indicator of nutritional status. As edema and hydration have the potential to affect weight and BMI values, therefore these factors will consider during the

determination of these parameters [35]. These parameters will be pictured on the chart to assess nutritional status.

**Intervention.** The most effective practice paradigm for the care and treatment of ASD is inter-professional teamwork employing a multidisciplinary approach [19]. Therefore, in this research researchers try to include the professionals who commonly work with ASD as a nutritionist, physiotherapists (PT), occupational therapists (OT), and speech and language therapists (SLT).

*Protocol for SD+RT group.* This group will receive treatments as structured dietary interventions consulted and maintained by a nutritionist. Regular therapy will include physiotherapy, occupational therapy, and speech and language therapy. As a common approach for both groups, conventional care from the teachers, parents, and counselors for ASD will be continued. Treatment dosage will depend on age, ASD symptoms, and severity.

Physiotherapy interventions include balance training as Land-based & swimming-based exercises [20], gait training to improve gross motor function and coordination of autism [21], coordination practice [36], hippotherapy or therapeutic horse riding [22], gross motor function practice [23], proprioception training to improve motor functions & improving lower extremity strength [24], aerobic exercise training to improve sleep, motor skill and mood among ASD children [25], muscle strengthening to improve higher executive function [26].

An expert speech and language therapist will provide treatment based on ASD symptoms. Common treatment will include- Applied Behavior Analysis (ABA) [37], Applied behavior Consequence (ABC) [38], Relationship Development Intervention (RDI) [39], Cognitive Behavior Intervention (CBI), Social communication intervention (e.g Social Story) [40], Picture Exchange, Communication System (PECS), Augmentative and alternative communication (AAC), Visual schedule [41].

Occupational therapy intervention includes sensory integration that includes the 10 key therapeutic strategies [42] in the fidelity tool, on the individual needs of each child as, (a) setting up the space to encourage interaction, (b) providing for physical security, (c) offering options for sensory input, (d) achieving and preserving appropriate arousal levels, (e) modifying activities to offer the ideal challenge, (f) assuring the success of activities, (g) providing direction for behavior self-regulation, (h) establishing a fun atmosphere, (I) choosing an activity together, and (j) encouraging therapeutic partnerships [43]. These 10 areas will be divided into 3 primary categories as (i) modifying the environment and giving sensory possibilities for the kid throughout the therapy session; (ii) encouraging adaptive responses and offering the ideal challenge; and (iii) building the therapist-child relationship [44]. Other interventions include gross motor function through exercise including pegboard games, dexterity drills, crossword puzzles, ball games, balance training, sensory integration exercises, and play therapy [45]. Moreover, community mobility or travel training [46], cognitive behavioral therapy [47, 48].

A structured diet (SD) will be designed by a nutritionist. Participants will provide with written instructions about the healthy, gluten-free, casein-free, soy-free diet and vitamin and mineral supplement [13]. The nutritionist will provide detailed advice. The diet plan will depend on age, nutritional status, and BMI. Other comorbid diseases and allergies of the child and adolescent to specific foods and physical activity levels will consider. The major guiding principles of the dietary plans will include, (1) Consuming enough fruit and vegetables, especially leafy greens (preferably whole fruit), (2) A sufficient amount and quality of protein, (3) A sufficient but moderate calorie intake, (4) Consuming fewer "junk" meals and substituting them with wholesome snacks, (5) A wholesome, gluten-free, casein-free, and soy-free diet (HGCSF) that includes foods like poultry, fish, and meat as well as fruits and vegetables, potatoes, rice, and baby rice cereal, and (6) Refrain from using synthetic flavors, colors, and preservatives.

Moreover, Vitamin or mineral supplements will be prescribed with the consent of a General Physician. The dosage will be determined by the physician.

*Protocol for UD+ RT group*. The group will receive Regular therapy as per the above protocol, but they will not receive any consultations from the nutritionist and will take the usual diet. After completion of the follow-up, they will be advised of the structured diet intervention based on the study result.

*Progression of interventions*. The intervention duration will be 12 weeks. 45 minutes of the session with the ASD Adolescents and their parents will be provided 3 times a week for each therapeutic intervention item. Participants will receive separate sessions for each therapy item (PT, OT, SLT, SD). The progression of interventions will be determined by subsequent professionals.

## Minimization of bias and blinding process

Assessors, patients, and therapists will be blinded to group allocation in this study. As this will be a multi-center trial of an organization, researchers plan to different assessors for each center. Randomization will be done by a person who is not involved in this research. The treatment providers will be blinded to the groups, but they will be aware of the treatment of their group, also participants will not be aware of the intervention of another group. The execution of the study will not involve the research team, and a separate assessor, treatment provider, monitoring team, and trial manager will be employed to guarantee trial coordination.

## Monitoring

The monitoring team will include 2 persons, who are not directly working on this trial. They will take responsibility for monitoring–intervention protocol, adverse effects, and enrollment of participants in groups. They will also audit data and carry out any interim analysis. They will report all kinds of changes in study methods and treatment to the Ethical Review Board (IRB) via the principal investigator.

## Safety measures and managing adverse effects

Although it is anticipated that the mentioned protocol won't result in any serious side effects, the monitoring team will take note of any unexpected occurrences that do happen during or after intervention and will principally inform the relevant specialists about them. The concerned therapist will make a note of it in their record and later let the lead investigator know. Before beginning, the child's parents will be asked about any food allergies. Any significant negative effects will be noted and published in the trial's final publication.

## Data analysis

According to the nature of the data, SPSS version 23 for Windows will be used for analysis. The distribution of data will be performed using normality indicators and normality tests such as the Kolmogorov-Smirnov test, and the Shapiro-Wilk test. Descriptive analysis will be performed by mean and standard deviation for the continuous data, and frequency and percentage categorical data. The baseline compatibility, between-group post-test, and follow-up variations among the groups will be assessed using MANOVA or the Multivariate Kruskal-Wallis (MKW) test, depending on the nature of the data. Repeated measure or Friedman's ANOVA Three measures will be analyzed within-group using an ANOVA, and additional post-hoc analyses will be performed using paired sample t-tests or Wilcoxon tests. We will perform the intention to treat the analysis process for the drop-out of cases taking more than one

month of interventions. In the case of a post hoc Bonferroni adjustment, the level of significance P < .05 will be converted to P < .0125.

## Ethical issues and informed consent

On October 18, 2022, the Institute of Physiotherapy Rehabilitation and Research (IPRR) of the Bangladesh Physiotherapy Association (BPA) granted the trial ethical approval (BPA-IPRR/IRB/10/18/3502). On November 25, 2022, the trial has been registered prospectively with the Clinical Trial Registry India (CTRI) (CTRI/2022/11/047653). The researcher will adhere to the Helsinki statement, to maintain ethical principles. The trial participants will sign a written informed consent before enrolment. The respondent's participation will be voluntary, and they will have the ability to leave at any point during the trial. Withdrawal from the trial will not make any effect on their treatment process. The completed trial dataset will be made available to the primary investigator, data auditors, and authors when the procedure has been made anonymous. The investigators are independent individuals who evaluate the patients and record their findings in hard copies; they are under contract not to disclose or have access to the final data. If there are any unfavorable consequences, there will be post-trial treatment.

## Study status

Participant recruitment for the study has started recruiting patients.

## Dissemination

After the end of the trial, a research paper will be submitted to an indexed journal for publication of the original research. Besides the trial, a seminar will be conducted to share the results with relevant stakeholders. A structured diet awareness program will also be organized if the study results are found to be effective. The trial results will be published as open access to ensure maximum visibility of the original research.

## Discussion

ASD is a neurological disease that manifests in early life and has a life-long impact [26]. ASD affects communication, behavior, education, personal and social life, even in a person's nutritional status [13]. This also places them at high risk for nutritional imbalances [13]. According to research, people with ASDs are nutritionally sensitive because they have a selective eating behavior and sensory sensitivity, which predisposes them to limited intakes [49]. Moreover, we hypothesized that a properly planned diet may improve autism symptoms and prevent happening of gastrointestinal disorders [50].

Rehabilitation through regular therapies has a positive outcome in ASD. Physiotherapy [20–26], Occupational therapy [42–48], Speech and language therapy [37–41], and nutritional consultation [13, 49, 50] are effective separately. Moreover, exercise has a direct impact on the stereotypical behavior of ASD [51].

This study fills up the research gap of the combined efficiency of nutritional and rehabilitation interventions for ASD. The result of the study will be contributing towards determining the appropriate comprehensive approach to the severity of autism, managing stereotypical behavior, and preventing anticipated health hazards induced by malnutrition for Adolescents with ASD. Also, this study can contribute to developing a comprehensive guidelines for ASD in Bangladesh. The strength of the study is representative sampling withdrawn from a sample frame determined through a population-based cross-sectional household survey and maintaining the rigor in methodology. Hence, with this sample size and study settings, the study

results are limited for the ASD residing in the Dhaka division in Bangladesh. Future multi-center trials in all of the eight divisions in Bangladesh might generate a result with greater external validity.

## Supporting information

**S1 File. Sample size calculation.**
(PDF)

**S2 File. Sample frame for the population-based screening.**
(PDF)

**S1 Protocol.**
(PDF)

## Acknowledgments

The authors acknowledge Ahamadullah Hil Galeb and Azharul Islam from the Department of Physiotherapy and Rehabilitation at the Jashore University of Science and Technology for their support in preparing the manuscript.

## Author Contributions

**Conceptualization:** Ruksana Akter, Nadia Afrin Urme, K. M. Amran Hossain, Tofajjal Hossain, Shameem Ahammad, Mst. Hosneara Yeasmin, Md. Zahid Hossain, Rashida Parvin, Md. Sohrab Hossain, Md. Ashrafuzzaman Zahid.

**Funding acquisition:** Ruksana Akter, Shameem Ahammad, Mst. Hosneara Yeasmin, Md. Zahid Hossain, Rashida Parvin, Md. Sohrab Hossain.

**Methodology:** Ruksana Akter, Nadia Afrin Urme, K. M. Amran Hossain, Tofajjal Hossain, Shameem Ahammad, Mst. Hosneara Yeasmin, Md. Zahid Hossain, Rashida Parvin, Md. Sohrab Hossain, Md. Ashrafuzzaman Zahid.

**Project administration:** Ruksana Akter, Nadia Afrin Urme, K. M. Amran Hossain, Tofajjal Hossain, Shameem Ahammad, Mst. Hosneara Yeasmin, Md. Zahid Hossain, Rashida Parvin, Md. Sohrab Hossain, Md. Ashrafuzzaman Zahid.

**Resources:** Ruksana Akter, Nadia Afrin Urme, K. M. Amran Hossain, Tofajjal Hossain, Shameem Ahammad, Mst. Hosneara Yeasmin, Md. Zahid Hossain, Rashida Parvin, Md. Sohrab Hossain, Md. Ashrafuzzaman Zahid.

**Software:** Ruksana Akter, Nadia Afrin Urme, K. M. Amran Hossain.

**Supervision:** Ruksana Akter, Nadia Afrin Urme, K. M. Amran Hossain, Tofajjal Hossain, Shameem Ahammad, Mst. Hosneara Yeasmin, Md. Zahid Hossain, Rashida Parvin, Md. Sohrab Hossain, Md. Ashrafuzzaman Zahid.

**Validation:** Ruksana Akter, Nadia Afrin Urme, K. M. Amran Hossain, Tofajjal Hossain, Shameem Ahammad, Mst. Hosneara Yeasmin, Md. Zahid Hossain, Rashida Parvin, Md. Sohrab Hossain, Md. Ashrafuzzaman Zahid.

**Visualization:** Ruksana Akter, Nadia Afrin Urme, K. M. Amran Hossain, Tofajjal Hossain, Shameem Ahammad, Mst. Hosneara Yeasmin, Md. Zahid Hossain, Rashida Parvin, Md. Sohrab Hossain, Md. Ashrafuzzaman Zahid.

**Writing – original draft:** Ruksana Akter, Nadia Afrin Urme, K. M. Amran Hossain, Tofajjal Hossain, Shameem Ahammad, Mst. Hosneara Yeasmin, Md. Zahid Hossain, Rashida Parvin, Md. Sohrab Hossain, Md. Ashrafuzzaman Zahid.

**Writing – review & editing:** Ruksana Akter, Nadia Afrin Urme, K. M. Amran Hossain, Tofajjal Hossain, Shameem Ahammad, Mst. Hosneara Yeasmin, Md. Zahid Hossain, Rashida Parvin, Md. Sohrab Hossain, Md. Ashrafuzzaman Zahid.

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
