## [Decision Letter · Decision Letter 0]

25 May 2023

PONE-D-23-06866Protocol for a Randomized Clinical Trial Comparing the Efficacy of Structured Diet (SD) and Regular Therapy (RT) for Children and Adolescents with Malnutrition having Autism Spectrum Disorder (ASD)PLOS ONE

Dear Dr. Zahid,

Thank you for submitting your manuscript to PLOS ONE. After careful consideration, we feel that it has merit but does not fully meet PLOS ONE’s publication criteria as it currently stands. Therefore, we invite you to submit a revised version of the manuscript that addresses the points raised during the review process.

Please note that we have only been able to secure a single reviewer to assess your manuscript. We are issuing a decision on your manuscript at this point to prevent further delays in the evaluation of your manuscript. Please be aware that the editor who handles your revised manuscript might find it necessary to invite additional reviewers to assess this work once the revised manuscript is submitted. However, we will aim to proceed on the basis of this single review if possible. 

We look forward to receiving your revised manuscript.

Kind regards,

Jianhong Zhou

Staff Editor

PLOS ONE

Additional Editor Comments:

1. Can you clearly stated please, how to evaluate or determine the socio-demographics related to malnutrition for ASD?

2. Can you defined the specific ASD types and their criteria which will be involved in the study?

3. Clearly state the reason why shall you select the sample only from one single center?

4. Did you calculated the sample size for this RCT? If yes then clearly state this.

5. You mentioned that you have received partial grant but why you did not acknowledge this?

6. You mention that you are doing study for Children and Adolescents with Malnutrition having Autism Spectrum Disorder (ASD) but why you selected 13 to 18 years aged participants.

Reviewers' comments:

Reviewer's Responses to Questions

**Comments to the Author**

1. Does the manuscript provide a valid rationale for the proposed study, with clearly identified and justified research questions?

Reviewer #1: Yes

2. Is the protocol technically sound and planned in a manner that will lead to a meaningful outcome and allow testing the stated hypotheses?

Reviewer #1: Partly

3. Is the methodology feasible and described in sufficient detail to allow the work to be replicable?

Reviewer #1: Yes

4. Have the authors described where all data underlying the findings will be made available when the study is complete?

Reviewer #1: Yes

5. Is the manuscript presented in an intelligible fashion and written in standard English?

Reviewer #1: Yes

6. Review Comments to the Author

You may also provide optional suggestions and comments to authors that they might find helpful in planning their study.

Reviewer #1: This is a study protocol to conduct a randomized controlled clinical trial to compare the effectiveness of combined RT + SD therapy, compared to UD, for children/adolescents with ASD. The study was registered within the Indian clinical trial registry , and was also approved by the respective Ethics board. The study objectives are on target, and the study design looks adequate. ASD is an important disease where more research is needed; hence, the study objectives are on target. However, I have some concerns/comments in the statistical design, analytical framework, etc, which may require closer attention:

1. Randomization: More details on the randomization procedure is needed; in particular, this reviewer thought that a blocked randomization would be helpful to achieve a 1:1 balanced recruitment within the 2 groups.

2. Outcome variables: Clearly state what will be measured, i.e., the outcome variables, both primary, and if necessary, the secondary. It was not clear, given that a bunch of variables to be collected has been mentioned.

3. Sample size/Power: The sample size/power writeup is half-hearted. In any randomized trial, it is recommended to power the study based on the primary outcome variable, using 5% significance, using a desired (moderate) effect size, and a specific statistical test, such as a 2-sample t-test, or a relevant nonparametric 2-sample test (if one suspects non-Gaussianity of the primary outcome/response variable). None of that has been specified; only what I see is 35 in each group (total of 70!).

4. Statistical Analysis: Looks adequate; any reasoning, why a formal mixed-effects modeling will not be conducted (controlling for possible covariates), compared to a repeated-measures ANOVA.

Also, Bonferroni adjustments are often harsh; I would recommend going with false discovery rates (FDRs).

5. Writing style:

The Discussion section should clearly state that the findings of this study would be only limited to the enrolled sample (which has a moderate sample size) of Bangladeshi subjects. The authors should allude to future trials with larger sample sizes (under medium to large effect sizes), and covering other populations, to establish the credibility of their findings.

7. PLOS authors have the option to publish the peer review history of their article (what does this mean?). If published, this will include your full peer review and any attached files.

Reviewer #1: No

---

## [Author Response · Author response to Decision Letter 0]

3 Jul 2023

PONE-D-23-06866

Protocol for a Randomized Clinical Trial Comparing the Efficacy of Structured Diet (SD) and Regular Therapy (RT) for Children and Adolescents with Malnutrition having Autism Spectrum Disorder (ASD)

PLOS ONE

Rebuttal letter to the Academic editor and reviewers 

Dear Academic editor 

We thank you for your contribution to improving the quality of the paper. In response to your invitation to submit a revised version of the manuscript, we have revised the manuscript addressing every comment. 

Response: We have addressed the files naming 

Response: We have revised and written details during the re-submission.

Response: Added ORCID IDs 

Additional Editor Comments:

1. Can you clearly stated please, how to evaluate or determine the socio-demographics related to malnutrition for ASD?

Response: First objective of the study was revised to determine the socio-demographics and anthropometric measures related to malnutrition for ASD in Bangladesh, The intended variables can be birth history, mother's health status during pregnancy (history of infection, trauma), living environment (urban or rural), family status (joint or single), parental status (separated or conjugal), caregiver (parent, family member or caregiver), living environment, attending school, attending rehab or dietary sessions, sports and amusement facility, age, gender, Body Mass Index. The previous studies found a relationship between these factors https://www.ncbi.nlm.nih.gov/pmc/articles/PMC4496734/

2. Can you defined the specific ASD types and their criteria which will be involved in the study?

Response: Details have been described in the diagnosis section. In the DSM-5, four previous types of Autism in DSM-IV (Autistic disorder, Asperger syndrome, childhood disintegrative disorder, and pervasive developmental disorder- not otherwise specified (PDD-NOS) were merged in an umbrella term Autism spectrum disorder (ASD). Also, the first diagnostic category of ASD indicates deficits in social communication and interaction. For an ASD diagnosis, all of the following symptoms must be present (1) impairments in socio-emotional engagement with another person, (2) problems in nonverbal behaviors during communication, (3) problems in making or continuing communication with others. The presence of two or more of the following symptoms associated with limited interest and repetitive behaviors is required for the second domain as (1) persistent or repetitive speech, movement, or holding an object, (2) adherence to routines, rituals of verbal or nonverbal behavior patterns, or excessive resistance to change, (3) abnormal fixated interests or intensity of focus, (4) more or less reactivity to sensory input, or (5) unusual interest in sensory aspects of the environment. According to the third category, symptoms must appear in infancy but may not completely develop until societal demands exceed capacity. We will not focus on any type according to DSM-IV, we will focus on the Autism spectrum according to DSM-V. 

3. Clearly state the reason why shall you select the sample only from one single center?

Response: Explained in the revised manuscript in the study setting section as “Two study centers of an organization (one for control and one for experimental) were selected because of the availability of a specialist multi-disciplinary team of physiotherapist, occupational therapist, speech and language therapist, child psychologist, educational psychologist, nutritionist, and special educators were available all together in that specialized organization.” 

4. Did you calculated the sample size for this RCT? If yes then clearly state this.

Response: Revised and clarified the sample size calculation as “Sample size calculation has been performed through the software ClinCalc [S1 File] estimating the key outcome as the score of the Gilliam Autism Rating Scale (GARS 2) second edition. 24,31 Sample size has been calculated as the anticipated minimal clinically important differences (MCID) of GARS-2 24 were estimated as 4.7±1.7 (0-10 Hiva scale converted from 0-126 GARS 2 score) with a baseline of 25% minimal clinical improvement, enrolment ratio 1:1, 80% power, and with the alpha value 0.05, the total sample stands as of 66. For safety, we will enroll 70 ASD adolescents with a minimum number of 35 participants in each group.” 

5. You mentioned that you have received partial grant but why you did not acknowledge this?

Response: We have revised the funding information section as “Partial funding of the trial was funded by the Bela Health and Education Foundation (BHEF), Grant number BHEF/2022/9/G2. The fund was specified for the identification and intervention provision for Adolescents with Autism Spectrum Disorder in Bangladesh. Publication or any other costs are self-funded by the authors.

6. You mention that you are doing study for Children and Adolescents with Malnutrition having Autism Spectrum Disorder (ASD) but why you selected 13 to 18 years aged participants. 

Response: Initially, we screened children and adolescents, but to keep the intervention protocol uniform for the participants in the means of rehabilitation and dietary interventions, we specified the ages 13-18 years. With your comments, we have revised the title and the manuscript as “Protocol for a Randomized Clinical Trial Comparing the Efficacy of Structured Diet (SD) and Regular Therapy (RT) for Adolescents with Malnutrition having Autism Spectrum Disorder (ASD)”. According to WHO, adolescent's age range is 10-19 years https://www.who.int/health-topics/adolescent-health#tab=tab_1

Response to Reviewer 

Dear Reviewer, 

We appreciate your time and scholarly comments to improve the quality of the manuscript. We have revised the manuscript and addressed all of your comments. 

Comments to the Author

1. Does the manuscript provide a valid rationale for the proposed study, with clearly identified and justified research questions?

Reviewer #1: Yes

2. Is the protocol technically sound and planned in a manner that will lead to a meaningful outcome and allow testing the stated hypotheses?

Reviewer #1: Partly

Response: We have revised to improve the manuscript 

3. Is the methodology feasible and described in sufficient detail to allow the work to be replicable?

Reviewer #1: Yes

4. Have the authors described where all data underlying the findings will be made available when the study is complete?

Reviewer #1: Yes

5. Is the manuscript presented in an intelligible fashion and written in standard English?

Reviewer #1: Yes

6. Review Comments to the Author

You may also provide optional suggestions and comments to authors that they might find helpful in planning their study.

Reviewer #1: This is a study protocol to conduct a randomized controlled clinical trial to compare the effectiveness of combined RT + SD therapy, compared to UD, for children/adolescents with ASD. The study was registered within the Indian clinical trial registry, and was also approved by the respective Ethics board. The study objectives are on target, and the study design looks adequate. ASD is an important disease where more research is needed; hence, the study objectives are on target. However, I have some concerns/comments in the statistical design, analytical framework, etc, which may require closer attention:

1. Randomization: More details on the randomization procedure is needed; in particular, this reviewer thought that a blocked randomization would be helpful to achieve a 1:1 balanced recruitment within the 2 groups.

Response: We have revised the section “Recruitment and randomization”. Also added the line “In the group allocation process, block randomization will be employed to ensure the balanced recruitment for two groups.” 

2. Outcome variables: Clearly state what will be measured, i.e., the outcome variables, both primary, and if necessary, the secondary. It was not clear, given that a bunch of variables to be collected has been mentioned.

Response: We have made two sections as explanatory variables (socio-demographic information and diagnosis of ASD) and primary outcome variables (GARS-2 and anthropometric measures) 

3. Sample size/Power: The sample size/power writeup is half-hearted. In any randomized trial, it is recommended to power the study based on the primary outcome variable, using 5% significance, using a desired (moderate) effect size, and a specific statistical test, such as a 2-sample t-test, or a relevant nonparametric 2-sample test (if one suspects non-Gaussianity of the primary outcome/response variable). None of that has been specified; only what I see is 35 in each group (total of 70!).

Response: Revised and clarified the sample size calculation as “Sample size calculation has been performed through the software ClinCalc [S1 File] estimating the key outcome as the score of the Gilliam Autism Rating Scale (GARS 2) second edition. 24,31 Sample size has been calculated as the anticipated minimal clinically important differences (MCID) of GARS-2 24 were estimated as 4.7±1.7 (0-10 Hiva scale converted from 0-126 GARS 2 score) with a baseline of 25% minimal clinical improvement, enrolment ratio 1:1, 80% power, and with the alpha value 0.05, the total sample stands as of 66. For safety, we will enroll 70 ASD adolescents with a minimum number of 35 participants in each group.”

4. Statistical Analysis: Looks adequate; any reasoning, why a formal mixed-effects modeling will not be conducted (controlling for possible covariates), compared to a repeated-measures ANOVA.

Also, Bonferroni adjustments are often harsh; I would recommend going with false discovery rates (FDRs).

Response: We assume no missing data in the post-test and follow-up, expect a normal distribution through block randomization, a simple post-hoc analysis (pre-post, post-follow-up), time as a category, and only two repetitions of data collection. All these factors suggest in favor of repeated-measures ANOVA. https://www.theanalysisfactor.com/six-differences-between-repeated-measures-anova-and-linear-mixed-models/

However, formal mixed-effects modeling, and false discovery rates both are great options. We thank the reviewers for the comment “Looks adequate” and for the suggestions, we can revise it until it is mandatory. 

5. Writing style:

The Discussion section should clearly state that the findings of this study would be only limited to the enrolled sample (which has a moderate sample size) of Bangladeshi subjects. The authors should allude to future trials with larger sample sizes (under medium to large effect sizes), and covering other populations, to establish the credibility of their findings.

Response: Revised in the discussion section between lines 365 and 368.

---

## [Decision Letter · Decision Letter 1]

18 Sep 2023

Protocol for a Randomized Clinical Trial Comparing the Efficacy of Structured Diet (SD) and Regular Therapy (RT) for Adolescents with Malnutrition having Autism Spectrum Disorder (ASD)

PONE-D-23-06866R1

Dear Dr. Zahid,

We’re pleased to inform you that your manuscript has been judged scientifically suitable for publication and will be formally accepted for publication once it meets all outstanding technical requirements.

Kind regards,

Ragab Kamal Elnaggar

Academic Editor

PLOS ONE

Additional Editor Comments (optional):

Reviewers' comments:

Reviewer's Responses to Questions

**Comments to the Author**

1. Does the manuscript provide a valid rationale for the proposed study, with clearly identified and justified research questions?

Reviewer #1: Yes

2. Is the protocol technically sound and planned in a manner that will lead to a meaningful outcome and allow testing the stated hypotheses?

Reviewer #1: Yes

3. Is the methodology feasible and described in sufficient detail to allow the work to be replicable?

Reviewer #1: Yes

4. Have the authors described where all data underlying the findings will be made available when the study is complete?

Reviewer #1: Yes

5. Is the manuscript presented in an intelligible fashion and written in standard English?

Reviewer #1: Yes

6. Review Comments to the Author

You may also provide optional suggestions and comments to authors that they might find helpful in planning their study.

Reviewer #1: The authors were able to address my previous concerns with a greater degree of satisfaction. I have no further comments.

7. PLOS authors have the option to publish the peer review history of their article (what does this mean?). If published, this will include your full peer review and any attached files.

Reviewer #1: No

---

## [Editor Report · Acceptance letter]

2 Oct 2023

PONE-D-23-06866R1 

Protocol for a Randomized Clinical Trial Comparing the Efficacy of Structured Diet (SD) and Regular Therapy (RT) for Adolescents with Malnutrition having Autism Spectrum Disorder (ASD) 

Dear Dr. Zahid:

I'm pleased to inform you that your manuscript has been deemed suitable for publication in PLOS ONE. Congratulations! Your manuscript is now with our production department. 

Kind regards, 

on behalf of

Professor Ragab Kamal Elnaggar 

Academic Editor

PLOS ONE